**Cite this article:** van Dijk LJA, Ehrlén J, Tack AJM. 2020 The timing and asymmetry of plant–pathogen–insect interactions. *Proc. R. Soc. B* **287**: 20201303.

ecology, plant science

compensatory growth, *Erysiphe alphitoides*, *Phalera bucephala*, plant–pathogen–insect interactions, SA- and JA-pathway, *Tuberculatus annulatus*

**Author for correspondence:**
Laura J. A. van Dijk
e-mail: laura.vandijk@su.se

# The timing and asymmetry of plant–pathogen–insect interactions

Laura J. A. van Dijk, Johan Ehrlén and Ayco J. M. Tack

Department of Ecology, Environment and Plant Sciences, Stockholm University, 106 91 Stockholm, Sweden

LJAvD, 0000-0003-1015-8496

Insects and pathogens frequently exploit the same host plant and can potentially impact each other's performance. However, studies on plant–pathogen–insect interactions have mainly focused on a fixed temporal setting or on a single interaction partner. In this study, we assessed the impact of time of attacker arrival on the outcome and symmetry of interactions between aphids (*Tuberculatus annulatus*), powdery mildew (*Erysiphe alphitoides*), and caterpillars (*Phalera bucephala*) feeding on pedunculate oak, *Quercus robur*, and explored how single versus multiple attackers affect oak performance. We used a multifactorial greenhouse experiment in which oak seedlings were infected with either zero, one, two, or three attackers, with the order of attacker arrival differing among treatments. The performances of all involved organisms were monitored throughout the experiment. Overall, attackers had a weak and inconsistent impact on plant performance. Interactions between attackers, when present, were asymmetric. For example, aphids performed worse, but powdery mildew performed better, when co-occurring. Order of arrival strongly affected the outcome of interactions, and early attackers modified the strength and direction of interactions between later-arriving attackers. Our study shows that interactions between plant attackers can be asymmetric, time-dependent, and species specific. This is likely to shape the ecology and evolution of plant–pathogen–insect interactions.

## 1. Introduction

In nature, it is the rule rather than the exception that multiple species of plant attackers, such as herbivores and pathogens, simultaneously or sequentially exploit the same host plant [1]. As a consequence, attackers might influence each other's performances, either directly or through induced plant responses [2,3]. Several studies have indeed shown that co-occurring insects and pathogens can have a major impact on each other's performance [4–6]. Despite increased attention to the interactions between plants, pathogens, and insects [7], our understanding of how attacker identity, order of arrival, and biotic context shape the outcome of plant–pathogen–insect interactions is still limited.

One might expect attackers to always negatively influence plant growth, both directly via damage and through inducing allocation of resources to plant defence mechanisms [8]. Still, reported effects on plant performance vary considerably [9–11], and this variation might be caused by multiple mechanisms. For example, plants can initiate compensatory growth after damage, thereby minimizing effects on growth [12,13]. The degree of plant tolerance is likely to differ among ontogenetic stages [14], especially for long-lived species where resource availability and allocation vary considerably during development [15]. Notably, the seedling stage is a critical life-history phase during which the impact of attackers can be especially pronounced, causing significant damage and high mortality [16]. The impact of attackers on plant growth and performance can also be modified when experiencing multiple attackers simultaneously [17]. As a consequence, the total effects can be

additive, synergistic (larger than the sum of individual effects), or antagonistic (smaller than the sum of individual effects) [2,9].

Attackers that share a host plant may not only affect the plant, but also induce plant responses that influence other attackers. The impact that attackers have on one another are not always in the same direction. In fact, asymmetry of attacker responses has been shown to be common among plant-feeding insects [18], and between insects and pathogens [6]. For example, beetle larvae (Plagiodera versicolora) showed decreased survival rates and slower development when feeding on willows infected with the rust fungus Melampsora allii-fragilis, while herbivore feeding promoted susceptibility of the host to fungal infection [19]. By contrast, bi-directional negative effects were found between the sap-feeding insect Bemisia tabaci and the biotrophic pathogen Odium neolycopersici [20]. Knowledge about the direction and symmetry of attacker responses is essential to understand attacker community dynamics and resulting consequences for the host plant. Despite this, the majority of studies to date that have investigated multiple-attacker scenarios focused only on the consequences for one of the attackers [5,21].

Attackers can impact one another even when their presence on the host plant is separated in time, with the first arriving attacker affecting later-arriving attackers [22,23]. Such priority effects can occur through niche pre-emption, meaning that the first attacker reduces resources available to the later-arriving attacker, or via niche modification, meaning that the first attacker alters the host niche, for example, by inducing an immune response. Two main plant defence pathways are the salicylic acid (SA) and the jasmonic acid (JA) pathway; the former is generally effective against biotrophic pathogens or sucking insects, whereas the latter protects the plant against necrotrophic pathogens or chewing insects [24]. Previous studies suggest reciprocal antagonism between the SA and JA pathways, whereby molecules that regulate the gene transcription for one pathway suppress gene transcription for the other pathway. This means that induction of the individual pathways is less efficient when both are induced at the same time [25]. While this defence antagonism allows a plant to prioritize defences and minimize the energetic costs [26], it also renders a plant more susceptible to subsequent attackers if these attackers are insensitive to the concurrent chemical pathway. Attackers can also modify host plant quality by changing the nutritional status and physiology [27]. For example, a fungal pathogen on birch positively influenced aphid performance by inducing the release of free amino acids in the phloem sap that benefitted aphid nutrition [28]. Research on priority effects for host–attacker interactions has hitherto mostly focused on closely related microbial species with high niche overlap [22], or on herbivore–herbivore interactions (e.g. [29–31]), while studies of the priority effects of plant–pathogen–insect interactions are still lacking. We also lack insights into the effects of early-arriving attackers on the interaction between later-arriving attackers.

The overarching aim of this study was to investigate the effects of simultaneously and sequentially arriving insect and pathogen attackers on the performance of tree seedlings and on each other's performance. For this, we used a multi-factorial growth chamber experiment in which we infested seedlings of the pedunculate oak Quercus robur with the caterpillar Phalera bucephala, the aphid Tuberculatus annulatus, and the biotrophic pathogen Erysiphe alphitoides. Oak seedlings were infested with zero, one, two, or three attacker species, with the order of attacker arrival differing among treatments. We addressed the following questions: (i) do single attackers affect plant performance? (ii) Is the impact of two attackers on plant performance additive, synergistic, or antagonistic? (iii) Do two attackers affect each other's performance when co-occurring on a host plant, and is the impact of attackers on each other's performance symmetric? (iv) Does the relative timing of the two attackers influence the outcome of the interaction? In the case of three attackers, can an early-arriving attacker alter the interaction outcome between two later-arriving attackers?

We expected that dual attack would result in either synergistic or antagonistic effects on plant performance, depending on the identity and combination of attacker species involved [9,17]. Specifically, the effects on plant performance would reflect positive, negative, or neutral effects of attackers on each other's performance, where the interaction outcome is expected to be mediated via crosstalk between plant defence pathways. Based on the results of previous studies, we expected aphids and mildew to induce the SA-pathway [32], and caterpillars to induce the JA-pathway. If indeed driven by defence crosstalk, attacker interactions are expected to be symmetric [25]. For example, mildew and aphids are expected to have a bi-directional negative impact on each other's performance due to shared sensitivity to SA, while caterpillars and aphids are expected to positively impact one another via inhibitive crosstalk between JA and SA. On the other hand, if other mechanisms are more important than defence crosstalk, such as resource competition, we might expect to find asymmetric interactions between attackers [33]. Finally, we predicted that early-arriving attackers would modify the interaction outcome between two late-arriving attackers, with the direction of this effect depending on which defence pathway was elicited by the first attacker. The a priori predictions based on defence crosstalk for each combination of attackers on plant and attacker performance are specified in electronic supplementary material, tables S2 and S4, respectively.

## 2. Material and methods

### (a) Study system

The pedunculate oak (Quercus robur) is a common deciduous tree in Europe, with its northernmost distribution reaching as far as central Norway and Sweden [34]. Oaks are a resource for a high diversity of phytophagous insects belonging to various feeding guilds [35], including sucking and chewing insects. Common oak aphids (Tuberculatus annulatus) are specialist insects, feeding on several species from the Quercus genus by sucking phloem sap from the leaf veins with their proboscis. Under favourable conditions, oak aphid populations can grow fast because females are viviparous and reproduction is asexual [36]. At the end of the growing season sexual reproduction takes over and oviparous wingless females lay eggs. The eggs hibernate during winter and hatch at the start of the next growing season. The caterpillars of the buff-tip moth (Phalera bucephala) feed on several deciduous tree species, such as oak and birch [37]. Females oviposit their eggs in clutches in June and July, and caterpillars feed on the leaves between July and early September. Fully developed caterpillars move into the soil to pupate, where the pupae hibernate. The adult moths emerge in spring.

Oaks also serve as a host to various pathogenic microbes. The oak powdery mildew *Erysiphe alphitoides* is a biotrophic fungus that commonly attacks the pedunculate oak [38,39]. During the growing season, the asexual spores are spread by the wind. Upon germination, the fungus grows epiphytically, with only the feeding organs penetrating the epidermal cells. At the end of the growing season, powdery mildew overwinters as sexual spores [40].

## (b) The experiment

We performed a multifactorial growth chamber experiment in which we studied organisms that are commonly co-occurring in European temperate ecosystems: the pedunculate oak, powdery mildew, common oak aphids, and the buff-tip moth [41–43]. The attacker combinations introduced on the oaks, and the timing thereof, reflected scenarios that were based on the phenology of the species involved and are likely to occur under natural conditions [44,45]. Furthermore, we strived to expose our seedlings to realistic attacker densities corresponding to the levels of attack and damage experienced by oak seedlings under natural conditions [45–47].

Acorns were collected in the autumn of 2017 in Stockholm (Sweden) and stored in a cold room. In spring 2018, acorns were categorized by weight to correct for maternal effects, planted in trays and placed in a climate chamber (L : D 10 : 14 h; 20 : 18°C). Acorn weight categories were: (i) less than 1 g, (ii) between 1 and 2 g, (iii) between 3 and 4 g, (iv) between 4 and 5 g, (v) between 5 and 6 g, and (vi) more than 6 g. Once germinated, acorns were transferred to individual pots (l × w × h: 7 × 7 × 18 cm) with potting soil (Krukväxtjord, SW Horto, Hammenhög, Sweden, with no additional nutrients added) and placed in trays with water to ensure a continuous water supply to the seedlings. Acorn weight categories were randomly distributed among the treatments described below.

Aphid colonies (*Tuberculatus annulatus*) originated from natural populations around Stockholm and were maintained for several generations in a climate chamber on oak seedlings prior to the experiment. Eggs of the buff-tip moth (*Phalera bucephala*) were ordered from Worldwide butterflies (UK). Powdery mildew colonies (*Erysiphe alphitoides*) originated from naturally occurring mildew colonies on oak trees in Stockholm and were maintained on oak seedlings in a climate chamber.

To investigate the impact of attacker identity, number of attacker species, and time of arrival on interaction outcome, we established 15 treatments consisting of different combinations of attacking organisms (figure 1). Each treatment was applied to 20 seedlings, with seedlings starting their treatment once all initial leaves were developed (*ca* 3 weeks after germination), with seedlings being randomly distributed among treatments. The 15 treatments can be categorized as belonging to one of five main types: (i) control seedlings (treatment 1), (ii) single attack (treatments 2–4), (iii) dual attack, with both attackers arriving on the plant at the same time (treatments 5–7), (iv) dual attack, with one attacker arriving earlier than the other (treatments 8–13), (v) triple attack, with one early-arriving attacker and two late-arriving attackers (treatments 14–15). We excluded the three-attacker treatment with mildew as the early-arriving attacker, because mildew could not be removed from the seedlings before the arrival of the late attackers. Importantly, attacker densities on the oak saplings during the experiment were reflective of densities encountered in the field [42,43,46].

Seedlings belonging to the different treatments (300 seedlings in total) were interspersed within a single climate chamber (L : D 10 : 14 h, 20°C : 18°C), and were covered by a pollination bag (PBS international, Scarborough, UK) to prevent the spread of attackers among oak seedlings. Each week, we measured seedling height, number of developed and undeveloped leaves, size of the largest leaf, and number of shoots.

To infect seedlings with powdery mildew (seedlings from nine treatments, thus 180 seedlings in total, figure 1), we gently brushed all leaves of the seedling with spores originating from a fully developed mildew colony (approx. 1 cm$^2$) [48,49]. Infected seedlings used for inoculation were maintained in the greenhouse. If visual signs of infection (i.e. small, whitish colonies) failed to establish within a week, leaves were inoculated again until infection was established. Differences in infection probability were likely caused by different levels of quantitative resistance [46,50]. As seedlings were randomized among treatments, we note that any effect of plant resistance would merely increase the residual error in the statistical models, and not affect estimates of among-treatment differences. To obtain a measure of mildew infection at the seedling level, the percentage of the total leaf surface of a seedling that was covered by mildew was visually estimated each week by the same person (L.J.A.v.D.). In the presence of chewing herbivory, mildew coverage was estimated for the remaining leaf tissues only.

To infest seedlings with aphids (seedlings from nine treatments, thus 180 seedlings in total, figure 1), we placed five wingless nymphs on a seedling, each on a separate leaf. To allow recording of aphid population growth during the entire experiment and to prevent extinctions or population growth estimates based on low numbers, we added new nymphs (up to five) if the population size decreased below three (needed for 39% of the seedlings after one week, but for less than 10% of the seedlings thereafter). Each week, we recorded the total number of aphids on the seedling, from which we calculated the population growth rate.

Caterpillars were placed on a seedling for one to two weeks (depending on caterpillar size) after which they were removed (on seedlings from nine treatments, thus 180 seedlings in total, figure 1). The caterpillars, unlike aphids and mildew, did thus not remain on the seedlings during the entire experiment. This incongruence in experimental set-up was unavoidable if we wanted to prevent complete defoliation (after one week the average defoliation was 26%, with a standard deviation of 27%). The effect of 'early-arriving caterpillars' should thus be interpreted as the effect of early chewing damage to seedlings, and the effect of 'co-occurring caterpillars' as the effect of chewing damage occurring simultaneously with the arrival of the other attacker. However, for simplicity, we refer to 'effect of early-arriving caterpillars' and 'effect of co-occurring caterpillars', respectively. For treatments 4, 7, and 12 (caterpillar only; caterpillar with aphids; and first aphid then caterpillar), the effect of aphids on short-term growth rate of the caterpillar was recorded. To test for the impact of mildew infection on long-term caterpillar development, we continued to feed caterpillars with mildew-infected or healthy leaves after removal of the caterpillar from the seedling. Throughout caterpillar development, we recorded larval instar and larval weight each week. To investigate the effect of treatment on development time and survival, we recorded pupation date and mortality.

## (c) Statistical analyses

Statistical analyses were performed in R v. 3.5.1 [51], using the packages *lme4* [52] and *emmeans* [53]. Significance of the models was assessed with the *Anova* function in the *car* package [54].

### (i) The impact of attackers on plant performance

To investigate the impact of treatment (i.e. the full set of attacker identities and combinations, including treatments 1–15; figure 1) on plant performance, we modelled plant height, the number of leaves, leaf size, and number of shoots as a function of treatment ($n = 300$ seedlings in each model). As we measured the same seedlings each week, we included the fixed effect date (factor) and the random effect seedling individual (linear mixed models as 'Repeated measures models', electronic supplementary

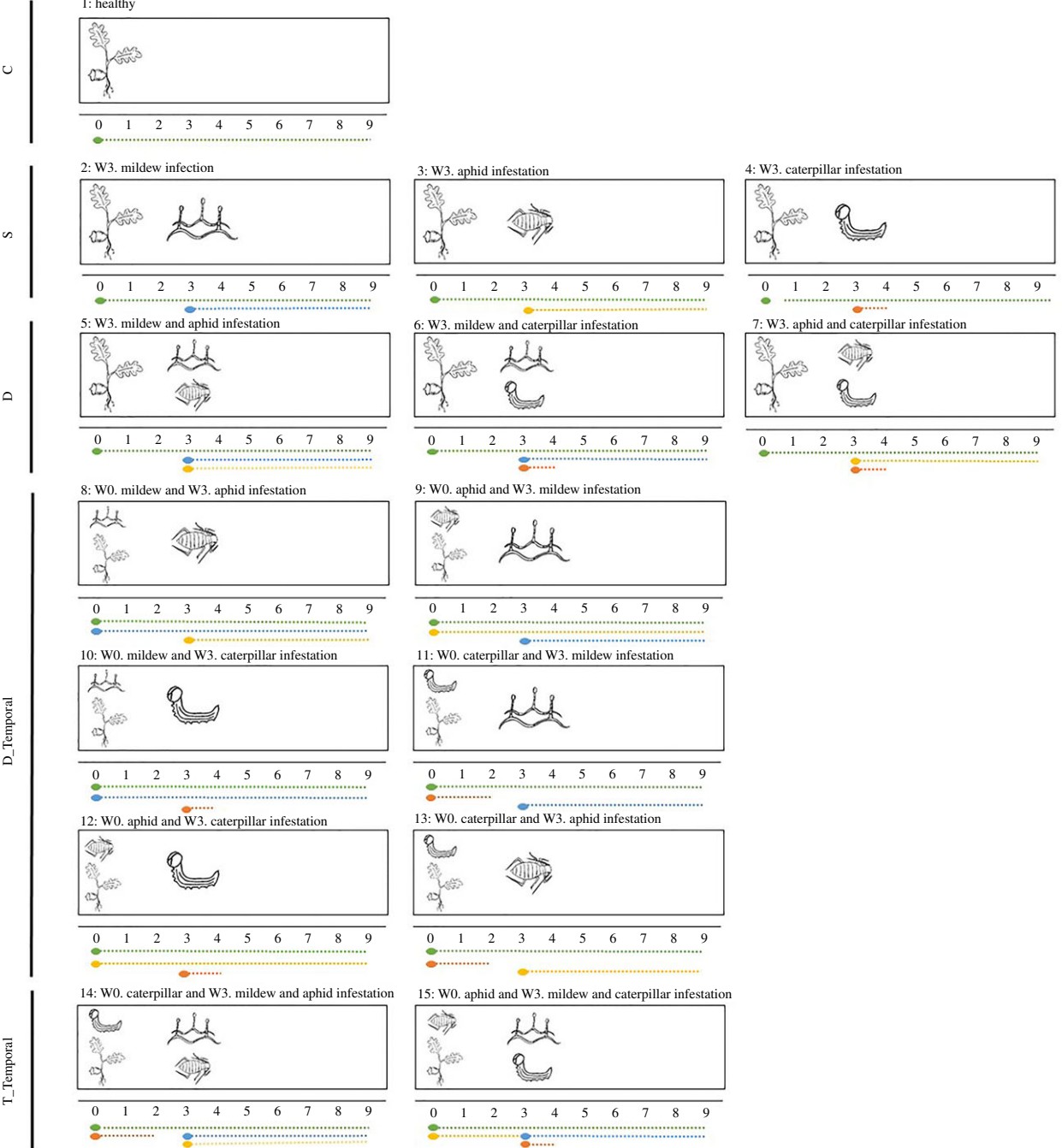

**Figure 1.** The experimental set-up, including 15 treatments. The weeks are numbered from 0 to 9. A dot means the organism enters the experiment, and a dotted line represents the time the organism is included in the experiment. Treatment types: C, Control; S, Single attack; D, Dual attack; D_Temporal, Dual attack, spatially separated; T_Temporal, triple attack, with one early attacker and two co-occurring late attackers. Green, oak seedling; blue, powdery mildew; yellow, aphid; orange, caterpillar. W0, week zero, indicating the start of the experiment, and W3, week 3. (Online version in colour.)

material, table S1). If needed, we transformed response variables to fit model assumptions (square root or log transformation, electronic supplementary material, table S1).

To explore the impact of each of the attacker species on plant performance, and to test whether co-occurring attackers had additive or non-additive effects on plant performance, we modelled plant performance as a function of mildew presence, aphid presence, caterpillar presence, date, and the interactions among all explanatory variables (linear mixed models as 'Additive risk models' [17], electronic supplementary material, table S2). Only treatments with healthy seedlings, seedlings with one attacker, and seedlings with two co-occurring attackers were included in these models (that is, treatments 1–7, $n = 180$ seedlings per model). The main effects mildew, aphids, and caterpillar showed whether the presence of a given attacker species affected plant performance. The interaction terms mildew × date, aphids × date, and caterpillar × date showed whether the impact on plant performance by an attacker varied over time. The interaction terms mildew × aphids, mildew × caterpillar, and aphids × caterpillar tested for additive effects of attackers (i.e. a non-significant interaction) versus non-additive effects of attackers (i.e. a significant interaction). Finally, the three-way interaction terms mildew × aphids × date, mildew × caterpillar × date, and aphids × caterpillar × date tested whether the effects of the interactions between mildew and aphids, mildew and caterpillars, and aphids and caterpillars varied over time.

To investigate the relative importance of maternal provisioning and plant attackers on plant growth, we included acorn size as a fixed continuous effect in all plant models. Also, the interaction between acorn size and treatment was included to check

whether the response of a plant to an attacker treatment was dependent on acorn size. As not all resources from the acorn are allocated to the seedling in the early phase of development, acorn size might be a better estimate of differences in plant resources and possibilities for future growth than initial plant size, and thus a more appropriate covariate in our models.

### (ii) The impact of attackers on each other's performance

To investigate the impact of attackers on each other's performance, we modelled mildew infection severity, aphid population size, and caterpillar growth (final instar weight, pupal weight, development time) and survival as a function of treatment. For mildew and aphids we further included date and the interaction date × treatment (linear mixed models as 'Repeated measures models', electronic supplementary material, table S3). For the models of caterpillar final and pupal weight, starting weight was added as a covariate. For the effect of aphids on caterpillar growth, we focused the analyses on the week where caterpillars and aphids were both present on the same plant (figure 1; electronic supplementary material, table S3).

To determine the impact of mildew, aphids, and caterpillars on each other, and the impact of an early-arriving attacker on the interaction outcome of later-arriving attackers for every week, we estimated treatment-specific contrasts with the function *emmeans* in the package *emmeans* (with Tukey adjustment for multiple comparisons, with seven treatment comparisons for six separate weeks). Together, these analyses answered the specific questions stated in electronic supplementary material, table S4.

## 3. Results

### (a) The impact of attackers on plant performance

Across all treatments, the impact of treatment (i.e. attacker identity and attacker combination) on plant performance was generally weak and inconsistent (electronic supplementary material, figure S1), with the effects varying through time (electronic supplementary material, table S5). The effects of attackers on plant performance were small compared to the positive effect of acorn size (electronic supplementary material, figure S2 and table S5), and acorn size did not influence the response of seedlings to attackers (electronic supplementary material, table S5). Mildew did not significantly affect any of the plant growth traits, and the effect of aphids and caterpillars on plant performance varied over time (electronic supplementary material, table S6). Chewing damage caused a drop in the number and size of leaves up to two weeks after caterpillar arrival, after which differences were no longer apparent (electronic supplementary material, figure S3b,c).

The effects of damage by mildew and caterpillars were additive during the entire experiment (electronic supplementary material, table S6; non-significant effect of the interaction between caterpillar presence, mildew presence, and date, on plant performance). By contrast, the additivity of the impact of mildew and aphids, and aphids and caterpillars, varied over time (electronic supplementary material, figure S2 and table S6; significant effect of the interaction between attackers and date, on plant performance). When present, non-additive effects were antagonistic, i.e. co-occurring attackers inflicted equal or less damage to the host plant than individual attackers (electronic supplementary material, figure S3; treatments with two attackers show equal or higher performance compared with treatments with only one attacker).

### (b) The impact of attackers on each other's performance

The interactions between co-occurring mildew, aphids, and caterpillars were either asymmetric or neutral (figure 4a). When co-occurring, aphids positively affected mildew performance, whereas mildew had a negative effect on aphids which increased in strength over time (figure 2a,c; electronic supplementary material, tables S7 and S8). Co-occurring caterpillars did not significantly influence mildew or aphid performance (figure 2b,d; electronic supplementary material, tables S7 and S8), and caterpillar growth rate was not significantly affected by co-occurring aphids (treatment 4 versus 7: $t = 0.91$, $p$-value = 0.64, electronic supplementary material, figure S4a). Caterpillars were, however, negatively affected by powdery mildew, experiencing a higher mortality (38% versus 21%, $\chi_1^2 = 5.96$, $p$-value = 0.015) and prolonged development times ($F_{1,152} = 14.31$, $p$-value < 0.001, electronic supplementary material, figure S4b,c) when feeding on leaves infected with mildew. Final instar weight and pupal weight were not significantly affected by treatment ($F_{1,124} = 0.04$, $p$-value = 0.84, and $F_{1,124} = 0.02$, $p$-value = 0.89, respectively).

The order of arrival of attackers had important effects on their performance (figure 4b). While mildew performance was promoted when it co-occurred with aphids, mildew performance was not significantly affected when aphids arrived earlier on the plant than mildew (figure 2a; electronic supplementary material, table S7). Likewise, the negative effect of mildew on aphid performance was no longer significant when mildew infected the seedling before the arrival of aphids. Conversely, caterpillars were unaffected when co-occurring with aphids, while there was a trend towards slower growth rates when feeding on plants previously infested by aphids (treatment 7 versus 12: $t = -2.39$, $p$-value = 0.05; electronic supplementary material, figure S4a).

The strength and direction of the interaction between two late-arriving, co-occurring attackers changed markedly in the presence of an early plant attacker. Early feeding by caterpillars weakened the positive effects of aphids on mildew infection, and counteracted the negative effect of mildew on aphid performance (figure 3a,c; electronic supplementary material, tables S7 and S8). Likewise, the outcome of the interaction between co-occurring caterpillars and mildew changed when aphids were present earlier. While mildew was unaffected by the co-occurrence of caterpillars, its performance was significantly impaired when aphids arrived earlier than the co-occurring mildew and caterpillars (figure 3b; electronic supplementary material, table S7).

## 4. Discussion

Our study showed that interactions among co-occurring attackers were either asymmetric or neutral. Interestingly, the order of arrival had a major impact on the strength and direction of the interaction. For example, aphids and mildew only affected each other when arriving at the same time, while caterpillar performance was only affected when aphids or mildew arrived before the caterpillar. Also, the interaction outcome between two attackers was modified when a third, early-arriving attacker was present on the plant. The impact of herbivores and pathogens on plant performance was generally weak and inconsistent. Overall, our findings illustrate that relative timing of attacker arrival shapes the outcome of plant–pathogen–insect interactions,

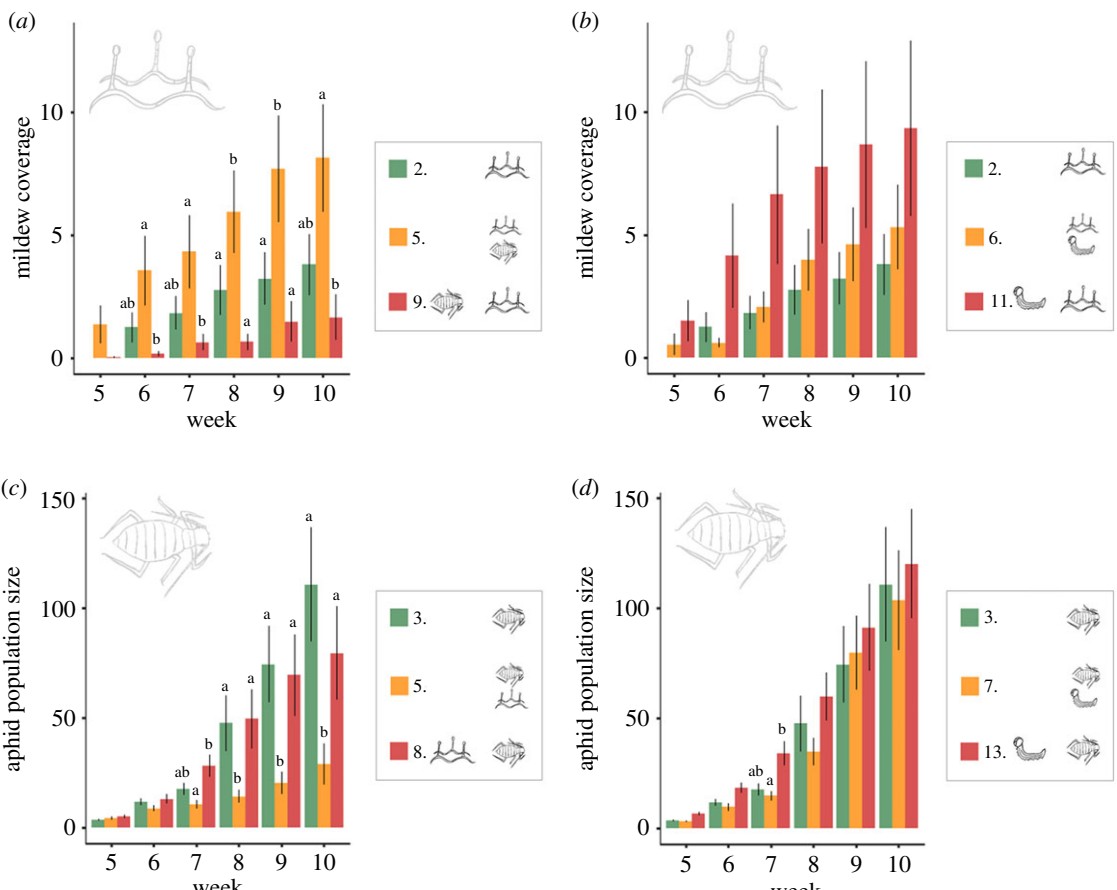

**Figure 2.** The effect of co-occurring attackers on mildew and aphid performance. (*a*) The mean mildew coverage when mildew is alone on a plant (2), when mildew is co-occurring with aphids (5), and when aphids arrive earlier than mildew (9). (*b*) The mean mildew coverage when mildew is alone on a plant (2), when mildew co-occurs with a caterpillar (6), and when the caterpillar arrives earlier than mildew (11). (*c*) The mean aphid population size when aphids are alone on a plant (3), when aphids co-occur with mildew (5), and when mildew arrives earlier than aphids (8). (*d*) The mean aphid population size when aphids are alone on a plant (3), when aphids co-occur with a caterpillar (7), and when the caterpillar arrives earlier than aphids (13). Each treatment had 20 replicates, and error bars present the standard errors. Significant differences between treatments in each week are indicated by letters above the bars. For statistical models and estimates, see electronic supplementary material, tables S3, S7, and S8. (Online version in colour.)

and that co-occurring attackers generally have asymmetric interactions (figure 4).

## (a) The impact of attackers on plant performance

We expected to find a negative impact of attackers on plant performance. Our findings instead suggested a general lack, or only weak effect, of aphids, mildew, and caterpillars, with the impact of aphids and caterpillars varying over time. Caterpillars reduced leaf number and size in the short term, but plants were able to regrow the lost tissue rapidly. After three weeks, size of caterpillar-attacked plants was again comparable to healthy plants. This finding indicates that plants experienced compensatory growth after caterpillar damage [12]. We did not detect any effect of mildew infection on seedling growth in our experiment, even though physiological impacts of mildew infection have been reported in an earlier study [55]. In contrast to the weak impact of attackers on plant performance, acorn size had a strong effect, with larger acorns producing larger seedlings. This finding indicates that maternal investment plays a vital role during the early life stages of a seedling, and adds to previous studies showing that acorn size is positively correlated to many fitness components [56]. Even though acorn resources are likely to enable compensatory growth after seedling damage [12,13,17], we did not find evidence that this ability was linked to acorn size, suggesting that the ability to compensate remains

proportional to plant growth for all sizes of acorns. Besides differences in the amount of stored resources, seedlings may also allocate resources to growth versus defence differently from older trees [15]. Energy allocation to growth is vital during early life stages, matching the observation that adult trees frequently have higher concentrations of defence chemicals in their leaves than seedlings [57,58]. Moreover, certain plant traits that can influence herbivores change during development, such as plant architecture, nutritional quality, and toughness of the leaves [59]. The impact of attackers on plant performance is thus likely to vary among ontogenetic stages, and our results, suggesting high tolerance towards attackers in seedlings, do not necessarily hold true for later stages of the life cycle. It is also true that attackers might have influenced root growth, which was not recorded in our study.

The impact of attackers on plant performance was mostly weak or absent, and the assessment of non-additivity should be interpreted with caution. However, for those attacker combinations where we found evidence of non-additive effects during some time periods, the combined effect was less severe than would be expected from the sum of their individual effects, i.e. effects were antagonistic. A recent meta-analysis found that on average, combined effects of attackers are additive, and that plant compensatory growth often overruled the synergistic effects of attackers [9]. For our results, it is likely that compensatory growth indeed explains why the effects of

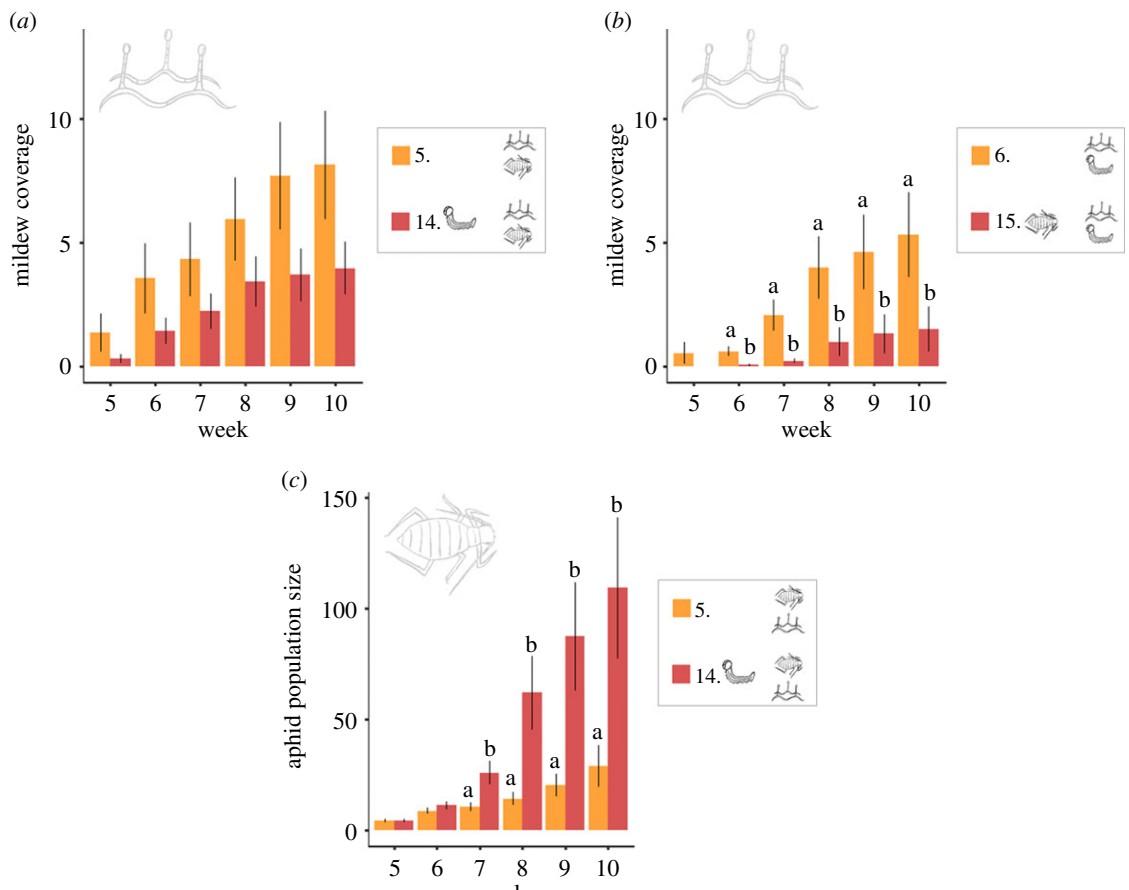

**Figure 3.** The effect of early-arriving attackers on the interaction between later-arriving, co-occurring attackers. (*a*) The mean mildew coverage when mildew co-occurs with aphids (5) and when caterpillars arrive earlier than the co-occurring mildew and aphids (14). (*b*) The mean mildew coverage when mildew co-occurs with a caterpillar (6) and when aphids arrive earlier than the co-occurring mildew and caterpillar (15). (*c*) The mean aphid population size when aphids co-occur with mildew (5) and when caterpillars arrive earlier than the co-occurring aphids and mildew (14). Each treatment had 20 replicates, and error bars present the standard errors. Significant differences between treatments in each week are indicated by letters above the bars. For statistical models and estimates, see electronic supplementary material, tables S3, S7, and S8. (Online version in colour.)

attackers on plant performance were absent or additive. On longer timescales though, compensatory growth can reduce growth due to energy reallocation from woody parts to leaves [60], and thus synergistic effects of co-occurring attackers may only become apparent after multiple years of attack. Thus, long-term studies are necessary to gain insights into delayed and cumulative effects of attacker damage.

## (b) The impact of attackers on each other's performance

The impact of co-occurring attackers on each other's performance was, when present, asymmetrical. Aphids positively impacted mildew, whereas mildew had a negative impact on aphids. For herbivorous insects, asymmetric interactions are commonly reported [18], and several examples exist for herbivore–pathogen interactions [19]. The underlying mechanism for asymmetric interactions between herbivores and pathogens cannot only be explained by defence pathway crosstalk: a simple SA–JA antagonism is expected to cause symmetric rather than asymmetric effects. A recent meta-analysis on the interactions between sequentially arriving attackers also found that reciprocal antagonism between the SA- and JA-pathway was not a universal trend, and thus, asymmetry between attacker responses can be expected [61]. Many other mechanisms might influence the outcome of interactions between multiple attackers. For example, aphids can act as vectors for fungal spores [62], and may thus aid in the spread of

disease within a seedling. A lack of impact of attackers on each other may occur when induced defence pathways are elicited at a very local scale, thereby being largely ineffective against subsequent attackers. For example, while the induction of SA is expected upon aphid attack, SA induction may occur only locally, i.e. at the feeding lesion [63], which may explain the lack of a significant negative influence of aphids on mildew via SA induction in our study. While it was not the aim of our study to distinguish local from distant interactions, we encourage future studies to explore the effect of contemporaneous SA and JA induction in nearby and distant plant tissues, and thereby resolve how spatial scale affects defence crosstalk [64] and shapes attacker interactions.

Priority effects, and the importance of order of attacker arrival, have been reported for diseases [22,65,66] and herbivore–herbivore interactions [23,29,30], though we lack insights when it comes to plant–pathogen–insect interactions in natural systems. Our study confirms that timing of arrival matters for the interaction outcome between attackers: mildew and aphid performances were only significantly affected when arriving simultaneously, while caterpillar performance was only impacted when leaves were previously attacked by mildew or aphids (the effect of aphids being marginally significant). It thus seems that, for aphids and mildew, direct or indirect short-term mechanisms played a more important role than indirect long-term mechanisms, whereas for caterpillars the opposite was true. Potentially, aphids could

(a) interaction outcome between co-occurring attackers

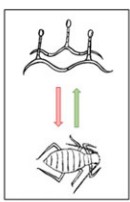
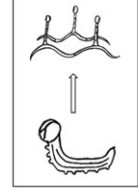

(b) the influence of an early attacker on a late attacker

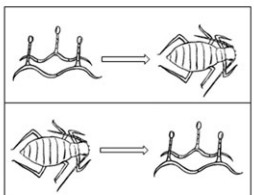
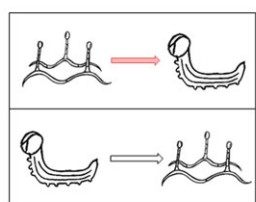
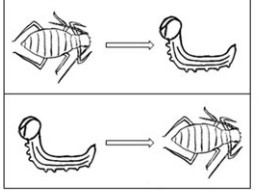

(c) the influence of an early attacker on the interaction outcome between two late attackers

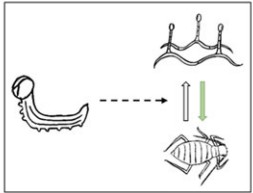
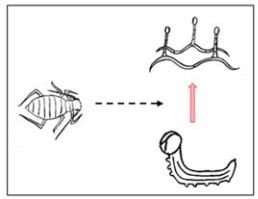

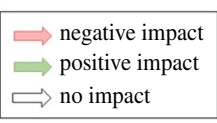
→ negative impact
→ positive impact
→ no impact

**Figure 4.** A schematic overview of the observed interactions among plant-attacking pathogens and insects. (a) The interaction outcome between two co-occurring attackers. (b) The influence of an early-arriving attacker on the performance of a late-arriving attacker. (c) The influence of an early-arriving attacker on the interaction outcome between two late-arriving attackers. (Online version in colour.)

have modified the host niche by manipulating plant physiology or nutritional status [63,67], reducing the plant quality for the later-arriving caterpillar. Mildew-infected leaves may have caused poorer caterpillar performance due to reduced photosynthetic activity [68], lower nutritional quality and water content [69], and tissue necrosis [41].

The strength and direction of the effect of two attackers on each other was strikingly different when a third, early-arriving attacker was present. Although aphids were negatively affected when co-occurring with mildew only, aphid performance was positively affected when a caterpillar arrived before the aphids and mildew. Moreover, mildew was positively impacted by aphids when arriving simultaneously, but this effect was weakened if caterpillars arrived before the mildew and aphids. Although previous studies have shown that early-arriving attackers may leave an imprint on the structure of the plant-associated community later during the season [23,70], we lack studies that investigate the impact of early-arriving attackers on the interaction outcome between later-arriving attackers. Even though we cannot generalize our findings at this point, our results suggest that early-arriving attackers can have important effects on interactions between later-arriving attackers.

In conclusion, this study demonstrates that plant–pathogen–insect interactions can be asymmetric, time-dependent and species-specific. This has important ecological and evolutionary consequences for plant-attacker communities. Given the predominance of asymmetric and neutral interactions between co-occurring pathogens and insects in this and other studies [6,19,61], we propose that asymmetry and neutrality is taken as a baseline prediction in future studies, despite its apparent incongruence with hypotheses derived from defence crosstalk that predict symmetry between attacker responses. The phenology of plants, insects, and pathogens is changing in response to increasing temperatures [71,72], including the number of generations per year [73], implying that the within-season order of arrival is likely to shift if the sensitivity to temperature differs among attackers. As suggested by the results of our study, such changes in relative timing are likely to have important effects on the structure of insect and microbial food webs.

Data accessibility. Data available from the Dryad Digital Repository: https://doi.org/10.5061/dryad.18931zctc [74].

Authors' contributions. L.J.A.v.D., A.J.M.T., and J.E. designed the research. L.J.A.v.D. conducted the experimental work, analysed the data, and wrote the manuscript, with contributions from all authors.

Competing interests. We declare we have no competing interests.

Funding. This research was supported by a grant from the Swedish Research Council Vetenskapsrådet (2015-03993 to A.J.M.T.).

Acknowledgements. We thank Giada Centenaro for her help with the experimental work.

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
