## [Reviewer comments · Proceedings of the Royal Society B: Biological Sciences]

Review History

RSPB-2020-1303.R0 (Original submission)

Review form: Reviewer 1

Recommendation

Accept with minor revision (please list in comments)

Scientific importance: Is the manuscript an original and important contribution to its field?
Excellent

General interest: Is the paper of sufficient general interest?
Excellent

Quality of the paper: Is the overall quality of the paper suitable?
Excellent

Is the length of the paper justified?
Yes

Should the paper be seen by a specialist statistical reviewer?
No

Do you have any concerns about statistical analyses in this paper? If so, please specify them explicitly in your report.

Yes

It is a condition of publication that authors make their supporting data, code and materials available - either as supplementary material or hosted in an external repository. Please rate, if applicable, the supporting data on the following criteria.

Is it accessible?

Yes

Is it clear?

Yes

Is it adequate?

Yes

Do you have any ethical concerns with this paper?

No

Comments to the Author

Line 61-62 specify which herbivores and fungi

Lines 187-188 specify how you standardised the application of inoculum of powdery mildew between plants, to ensure all plants received the same amount - makes statement in line 190-191 justified.

Lines 226-227 How did you account for differing numbers of leaves/shoots on plants at the beginning of the treatment period? I.e. why not instead measure growth rate as fixed effect?

Lines 225-245 it is not quite clear to me why these two sets of models were used. Why was it necessary to test plant performance as a function of treatment rather than the second set of model where attacker identity was tested? Isn't this basically repeating the same analysis? Also, there are a very large number of explanatory variables being tested here (I count 16 between the two sets of models explained) for 180 (?) observations, as well as multiple comparisons testing, which reduces statistical power. I wonder why so many metrics of plant performance were tested - could you just use one and thereby reduce the overall number of variables tested? Three way interaction terms are also very difficult to interpret. It might have been better to first explore the data using exploratory models and plots before deciding to fit all of the models. In any case, the total number of models fitted and the sample size needs to be quoted for each model, and the number of tests of multiple comparisons of means also needs to be recorded. What method of model selection / simplification was used? This should also be discussed.

Line 230 square root, not squared-root

General comment on figures: The ranges of the boxplots in the supp figures need to be explained. Boxplots should have a dot for means as well as medians shown, as medians are not always the best way to explore differences between treatment levels.

Line 290: But looking at figure 2a and as you state further down the results, isn't it the case that aphids only improved mildew performance when they were cooccurring, not when aphids arrived before mildew?

Lines 382-383: The suggestion that aphids induced the salicylic acid only locally...which pathway do they normally induce? My understanding is that biotrophic pathogens also induce the SA pathway, so the antagonistic hypothesis proposed would then make sense. Please clarify in the text which pathway you think each of the attackers induce.

Review form: Reviewer 2 (Bastien Castagneyrol)

Recommendation

Accept with minor revision (please list in comments)

Scientific importance: Is the manuscript an original and important contribution to its field?

Excellent

General interest: Is the paper of sufficient general interest?

Good

Quality of the paper: Is the overall quality of the paper suitable?

Excellent

Is the length of the paper justified?

Yes

Should the paper be seen by a specialist statistical reviewer?

No

Do you have any concerns about statistical analyses in this paper? If so, please specify them explicitly in your report.

No

It is a condition of publication that authors make their supporting data, code and materials available - either as supplementary material or hosted in an external repository. Please rate, if applicable, the supporting data on the following criteria.

Is it accessible?

No

Is it clear?

N/A

Is it adequate?

N/A

Do you have any ethical concerns with this paper?

No

Comments to the Author

Dear Editor,

In their paper « The timing and asymmetry of plant-pathogen-insect interactions », the authors address the effect of single vs. multiple attacks on plant growth as well as simultaneous vs. delayed interactions among attackers. They show that (1) pathogen and herbivore attack have weak or inconsistent effects on plant growth and that (2), when there are significant interactions among attackers, they are generally antagonistic and depend on the timing of the interaction. The most interesting result (for me) is that initial attack by a given attacker modifies the strength and direction of interactions between other attackers. The results are a bit complex and it is hard to navigate from the main text to the numerous supplementary materials, but I guess this is because word length limitation. The authors handled this fairly well. Overall, I liked the paper very much, because it is clear and well written, and because this is exactly the type of research I would have loved conducting. I only have a few comments/suggestions.

All the best,
Bastien Castagneyrol (I am fine with signing my comments)

@authors:

Paragraph starting L42 presents first the « positive » effect of attackers on plants, then the « negative » effects, but there is no clear transition between both, which makes it hard to follow. In addition, most of references are about herbivores. Any chance you can add references on pathogens as well.

L64 « ...understand community dynamics » → AND resulting consequences on the host plant ?

L104 - This prediction is a bit vague (« synergistic or antagonistic »). Do you expect that some attackers would have more important consequences on plant growth ? Could you have a more precise prediction here ? Maybe not.

L125 (and elsewhere) is « family » really needed ?

L136 - It took me some time to realise that caterpillars may attack quite late in the season, after oak leaves have been colonized by mildew. This is because in other places, several defoliating species attack earlier in the season, at the same time as budburst. In this case, the interaction between mildew and caterpillars is more unidirectional. I think this could be more explicit here, for it can be confusing. Also, I wondered how common is *Bucephala* in the study area and whether the results would have been different with another species.

L140 - There are two cryptic mildew species, right ? How can you be sure yours is *E. alphitoides* ?

L152-158 - Any information about nutrients and watering treatment ?

L167 - « number of attackers » is unclear. This is the number of different attackers, not the density of a given attacker (that would have been another, really interesting question too).

L169-175 - That would be nice to give the number of treatments for each of these five groups of treatments.

L181 - Were seedlings bagged individually ? If so, then I don't understand the comment L206 (and following) suggesting that caterpillars were free to move across seedlings.

L193 - Duplicated « the »

L194 - did you estimate the % leaf area covered by mildew or did you give an overall attack score at seedling level ? Is that representative of mildew infection ? What if some leaf area was missing because of herbivory ? How was it taken into account ?

L217 - Because some caterpillars feed at night, or only during the day, the amount of food in the gut may vary with time, which means that there is a risk larva weight is not the same if weighted say at 9am or 4pm. Usually, people keep larvae for a few hours without food to avoid this bias. Not sure this is a big deal here. What do you think ?

L228 - Why was time fitted as a factor. There are 6 different levels. I would have used it as a continuous variable. The degrees of freedom would have been lower and it would have been easier to interpret the interaction term, as it would have been the slope. Here, with no indication on pairwise differences among time × treatment interactions, it is hard to figure out what it means.

L242 – For additivity vs. non additivity, I had to write model equation to be convinced that the interaction term represents the non-additive effect. This is not absolutely needed but you may want to consider writing the equation. Just to be sure I understood properly, with e.g. A1 and A0 presence/absence of aphids, and M1/M0 presence/absence of mildew, and Y the predicted response variable:

$$Y = \beta_{_0} + \beta_{_1} \times A1 + \beta_{_2} \times M1 + \beta_{_3} \times A1 \times M1$$

with $\beta_{_0}$ the intercept (corresponding to predictions for A0 and M0). If $\beta_{_3} = 0$ (i.e., no interaction), then Y in A1-M1 is simply predicted from A1 only, M1 only and $\beta_{_0}$, $\beta_{_1}$, $\beta_{_2}$, is that correct?

I realized that models are provided in supplementary tables. I am fine with the notation, but some people complain that what you report is not model equation, strictly speaking. There is nothing to do with this comment, just keep in mind that sometimes it helps having the model equation(s) properly written.

L252 – When analyzing growth rate, it is usually suggested to use initial weight as a covariate and to interpret the initial weight \times treatment interaction as growth rate. Here, you only mention « final instar weight », while initial weight was variable across individuals.

Discussion

L273 – Can you precise whether the effect of acorn size was positive or negative ?

L298 – Treatment, not diet

L331 – You only focused on above ground growth. It is possible that treatments had a strong influence on belowground growth. This is particularly important for seedlings. Also, the effects of treatments on oak seedlings may have been delayed. Is there any chance you looked at growth the next year ?

Also, about growth, is it possible that treatments were not « damaging » enough ? I would have reported mildew infection, % herbivory and aphid load in the 'Results' section and discuss whether these scores are high/low and representative or not of what could be seen in the wild.

L379 – was the effect of aphid « wound » on mildew really relevant ? I suspect that the mechanical effect of leaf chewing by caterpillars is more significative.

L385 – is there any chance you looked at interactions at the level of individual leaves ? Were interactions local or only distant ? That would help discussing the effect of local vs systemic defense induction

Figure 4 helps a lot. But : (1) could you make arrow width proportional to model coefficient (1bis) avoid grey vs. red; and (2) I like cartoons but I am not a big fan of having realistic vs. cartoonish drawings in the same figure.

Figures – That would be nice to have the results of contrast analysis (where appropriate) indicating which comparisons are significant or not. Also, I did not really understand why in some instance you have bvargraphs and, elsewhere, boxplots or lines. OK, lines better show some continuity in the response variable (e.g., seedling measurements), but then, you could have used it for mildew coverage and aphid population size as well, for the same reason.

It is a bit frustrating the title does not reflect very well the fact that you also address the effect of single vs. multiple attacks on the plant.

Otherwise, this is a very nice paper :)

Decision letter (RSPB-2020-1303.R0)

14-Jul-2020

Dear Miss van Dijk:

Your manuscript has now been peer reviewed and the reviews have been assessed by an Associate Editor. The reviewers' comments (not including confidential comments to the Editor) and the comments from the Associate Editor are included at the end of this email for your reference. As you will see, the reviewers and the Editors have raised some concerns with your manuscript and we would like to invite you to revise your manuscript to address them.

Research ethics:

Use of animals and field studies:

It is a condition of publication that you make available the data and research materials supporting the results in the article. Please see our Data Sharing Policies

(<https://royalsociety.org/journals/authors/author-guidelines/#data>). Datasets should be deposited in an appropriate publicly available repository and details of the associated accession number, link or DOI to the datasets must be included in the Data Accessibility section of the article (<https://royalsociety.org/journals/ethics-policies/data-sharing-mining/>). Reference(s) to datasets should also be included in the reference list of the article with DOIs (where available).

If you wish to submit your data to Dryad (<http://datadryad.org/>) and have not already done so you can submit your data via this link [http://datadryad.org/submit?journalID=RSPB&manu=\(Document not available\)](http://datadryad.org/submit?journalID=RSPB&manu=(Document%20not%20available)), which will take you to your unique entry in the Dryad repository.

Please submit a copy of your revised paper within three weeks. If we do not hear from you within this time your manuscript will be rejected. If you are unable to meet this deadline please let us know as soon as possible, as we may be able to grant a short extension.

Best wishes,
Professor Gary Carvalho
<mailto:proceedingsb@royalsociety.org>

Associate Editor
Board Member: 1
Comments to Author:

Both reviewers recommend accepting the manuscript for publication I am very happy to agree that in terms of relevance, timeliness, the experimental design, etc. the study should be published by Proc Roy Soc.

It is also true that there quite a few and detailed comments. I would therefore recommend for the authors to read and consider the comments carefully. I'm sure it will make for a stronger paper. Given an adequate response, I'd be happy to recommend the manuscript for publication.

Reviewer(s)' Comments to Author:

Referee: 1

Comments to the Author(s)

Line 61-62 specify which herbivores and fungi

Lines 187-188 specify how you standardised the application of inoculum of powdery mildew between plants, to ensure all plants received the same amount - makes statement in line 190-191 justified.

Lines 226-227 How did you account for differing numbers of leaves/shoots on plants at the beginning of the treatment period? I.e. why not instead measure growth rate as fixed effect?

Lines 225-245 it is not quite clear to me why these two sets of models were used. Why was it necessary to test plant performance as a function of treatment rather than the second set of model where attacker identity was tested? Isn't this basically repeating the same analysis? Also, there are a very large number of explanatory variables being tested here (I count 16 between the two sets of models explained) for 180 (?) observations, as well as multiple comparisons testing, which reduces statistical power. I wonder why so many metrics of plant performance were tested - could you just use one and thereby reduce the overall number of variables tested? Three way interaction terms are also very difficult to interpret. It might have been better to first explore the data using exploratory models and plots before deciding to fit all of the models. In any case, the total number of models fitted and the sample size needs to be quoted for each model, and the number of tests of multiple comparisons of means also needs to be recorded. What method of model selection / simplification was used? This should also be discussed.

Line 230 square root, not squared-root

General comment on figures: The ranges of the boxplots in the supp figures need to be explained. Boxplots should have a dot for means as well as medians shown, as medians are not always the best way to explore differences between treatment levels.

Line 290: But looking at figure 2a and as you state further down the results, isn't it the case that aphids only improved mildew performance when they were cooccurring, not when aphids arrived before mildew?

Lines 382-383: The suggestion that aphids induced the salicylic acid only locally...which pathway do they normally induce? My understanding is that biotrophic pathogens also induce the SA pathway, so the antagonistic hypothesis proposed would then make sense. Please clarify in the text which pathway you think each of the attackers induce.

Referee: 2

Comments to the Author(s)

Dear Editor,

In their paper « The timing and asymmetry of plant-pathogen-insect interactions », the authors address the effect of single vs. multiple attacks on plant growth as well as simultaneous vs. delayed interactions among attackers. They show that (1) pathogen and herbivore attack have weak or inconsistent effects on plant growth and that (2), when there are significant interactions among attackers, they are generally antagonistic and depend on the timing of the interaction. The most interesting result (for me) is that initial attack by a given attacker modifies the strength and direction of interactions between other attackers. The results are a bit complex and it is hard to navigate from the main text to the numerous supplementary materials, but I guess this is because word length limitation. The authors handled this fairly well. Overall, I liked the paper very much, because it is clear and well written, and because this is exactly the type of research I would have loved conducting. I only have a few comments/suggestions.

All the best,

Bastien Castagnayrol (I am fine with signing my comments)

@authors:

Paragraph starting L42 presents first the « positive » effect of attackers on plants, then the « negative » effects, but there is no clear transition between both, which makes it hard to follow. In addition, most of references are about herbivores. Any chance you can add references on pathogens as well.

L64 « ...understand community dynamics » → AND resulting consequences on the host plant ?

L104 - This prediction is a bit vague (« synergistic or antagonistic »). Do you expect that some attackers would have more important consequences on plant growth ? Could you have a more precise prediction here ? Maybe not.

L125 (and elsewhere) is « family » really needed ?

L136 - It took me some time to realise that caterpillars may attack quite late in the season, after oak leaves have been colonized by mildew. This is because in other places, several defoliating species attack earlier in the season, at the same time as budburst. In this case, the interaction between mildew and caterpillars is more unidirectional. I think this could be more explicit here, for it can be confusing. Also, I wondered how common is *Bucephala* in the study area and whether the results would have been different with another species.

L140 - There are two cryptic mildew species, right ? How can you be sure yours is *E. alphitoides* ?

L152-158 - Any information about nutrients and watering treatment ?

L167 - « number of attackers » is unclear. This is the number of different attackers, not the density of a given attacker (that would have been another, really interesting question too).

L169-175 - That would be nice to give the number of treatments for each of these five groups of treatments.

L181 - Were seedlings bagged individually ? If so, then I don't understand the comment L206 (and following) suggesting that caterpillars were free to move across seedlings.

L193 - Duplicated « the »

L194 - did you estimate the % leaf area covered by mildew or did you give an overall attack score at seedling level ? Is that representative of mildew infection ? What if some leaf area was missing because of herbivory ? How was it taken into account ?

L217 - Because some caterpillars feed at night, or only during the day, the amount of food in the gut may vary with time, which means that there is a risk larva weight is not the same if weighted say at 9am or 4pm. Usually, people keep larvae for a few hours without food to avoid this bias. Not sure this is a big deal here. What do you think ?

L228 - Why was time fitted as a factor. There are 6 different levels. I would have used it as a continuous variable. The degrees of freedom would have been lower and it would have been easier to interpret the interaction term, as it would have been the slope. Here, with no indication on pairwise differences among time × treatment interactions, it is hard to figure out what it means.

L242 - For additivity vs. non additivity, I had to write model equation to be convinced that the interaction term represents the non-additive effect. This is not absolutely needed but you may want to consider writing the equation. Just to be sure I understood properly, with e.g. A1 and A0

presence/absence of aphids, and M1/M0 presence/absence of mildew, and Y the predicted response variable:

$$Y = \beta_0 + \beta_1 \times A1 + \beta_2 \times M1 + \beta_3 \times A1 \times M1$$

with β_0 the intercept (corresponding to predictions for A0 and M0). If $\beta_3 = 0$ (i.e., no interaction), then Y in A1-M1 is simply predicted from A1 only, M1 only and β_0 , β_1 , β_2 , is that correct?

I realized that models are provided in supplementary tables. I am fine with the notation, but some people complain that what you report is not model equation, strictly speaking. There is nothing to do with this comment, just keep in mind that sometimes it helps having the model equation(s) properly written.

L252 - When analyzing growth rate, it is usually suggested to use initial weight as a covariate and to interpret the initial weight \times treatment interaction as growth rate. Here, you only mention « final instar weight », while initial weight was variable across individuals.

Discussion

L273 - Can you precise whether the effect of acorn size was positive or negative ?

L298 - Treatment, not diet

L331 - You only focused on above ground growth. It is possible that treatments had a strong influence on belowground growth. This is particularly important for seedlings. Also, the effects of treatments on oak seedlings may have been delayed. Is there any chance you looked at growth the next year ?

Also, about growth, is it possible that treatments were not « damaging » enough ? I would have reported mildew infection, % herbivory and aphid load in the 'Results' section and discuss whether these scores are high/low and representative or not of what could be seen in the wild.

L379 - was the effect of aphid « wound » on mildew really relevant ? I suspect that the mechanical effect of leaf chewing by caterpillars is more significant.

L385 - is there any chance you looked at interactions at the level of individual leaves ? Were interactions local or only distant ? That would help discussing the effect of local vs systemic defense induction

Figure 4 helps a lot. But : (1) could you make arrow width proportional to model coefficient (1bis) avoid grey vs. red; and (2) I like cartoons but I am not a big fan of having realistic vs. cartoonish drawings in the same figure.

Figures - That would be nice to have the results of contrast analysis (where appropriate) indicating which comparisons are significant or not. Also, I did not really understand why in some instance you have bvargraphs and, elsewhere, boxplots or lines. OK, lines better show some continuity in the response variable (e.g., seedling measurements), but then, you could have used it for mildew coverage and aphid population size as well, for the same reason.

It is a bit frustrating the title does not reflect very well the fact that you also address the effect of single vs. multiple attacks on the plant.

Otherwise, this is a very nice paper :)

Author's Response to Decision Letter for (RSPB-2020-1303.R0)

See Appendix A.

Decision letter (RSPB-2020-1303.R1)

01-Sep-2020

Dear Miss van Dijk

I am pleased to inform you that your manuscript entitled "The timing and asymmetry of plant-pathogen-insect interactions" has been accepted for publication in Proceedings B.

Open Access

Paper charges

Sincerely,

Professor Gary Carvalho

Associate Editor:

Board Member

Comments to Author:

I suggest the authors account adequately for the comments by referees made on the last version, which both referees recommended for publication. It's a stronger manuscript for the changes.

Well done to the authors.

Appendix A

Editor and Reviewer comments are shown in black font and author answers in green font. Author responses are numbered as R1-R34. A copy of the revised manuscript including tracked changes is provided in the end of this document.

Subject Editor

Both reviewers recommend accepting the manuscript for publication I am very happy to agree that in terms of relevance, timeliness, the experimental design, etc. the study should be published by Proc Roy Soc.

It is also true that there quite a few and detailed comments. I would therefore recommend for the authors to read and consider the comments carefully. I'm sure it will make for a stronger paper. Given an adequate response, I'd be happy to recommend the manuscript for publication.

R1. We are happy to hear that the subject editor and reviewers appreciated our manuscript, and have now carefully addressed all reviewer comments. We would like to thank the reviewers for their detailed and insightful comments that greatly helped us to further improve our manuscript.

Reviewer 1

Line 61-62 specify which herbivores and fungi

R2. As suggested, we now give the Latin binomials of the rust fungus and beetle larvae (lines 61-62). We also added the Latin binomials for the species in the subsequent example (lines 64-65).

Lines 187-188 specify how you standardised the application of inoculum of powdery mildew between plants, to ensure all plants received the same amount - makes statement in line 190-191 justified.

R3. We now clarified the methods used for the application of mildew spores to our seedlings in lines 196-199 (*"To infect seedlings with powdery mildew (seedlings from 9 treatments, thus 180 seedlings in total, Fig. 1), we gently brushed all leaves of the seedling with spores originating from a fully developed mildew colony (approx. 1 cm²) (Nicot et al., 2002; Mursinoff & Tack, 2017). Infected seedlings used for inoculation were maintained in the greenhouse."*) We carefully followed the inoculation methods used in Mursinoff and Tack (2017) and Nicot et al. (2002), to whom we now also refer in this sentence.

Lines 226-227 How did you account for differing numbers of leaves/shoots on plants at the beginning of the treatment period? I.e. why not instead measure growth rate as fixed effect?

R4. This is a good point, and we agree that the reasons for our choice of covariates should be clearer when describing the statistical methods. Both acorn size and initial plant size are relevant options, though we had to pick one of these since they were highly correlated. We now added a statement to the statistical section (lines 268-271) to explain why acorn size, and not initial plant size, was added as a covariate to the model (*“As not all resources from the acorn are allocated to the seedling in the early phase of development, acorn size might be a better estimate of differences in plant resources and possibilities for future growth than initial plant size, and thus a more appropriate covariate in our models.”*).

Lines 225-245 it is not quite clear to me why these two sets of models were used. Why was it necessary to test plant performance as a function of treatment rather than the second set of model where attacker identity was tested? Isn't this basically repeating the same analysis? Also, there are a very large number of explanatory variables being tested here (I count 16 between the two sets of models explained) for 180 (?) observations, as well as multiple comparisons testing, which reduces statistical power. I wonder why so many metrics of plant performance were tested - could you just use one and thereby reduce the overall number of variables tested? Three way interaction terms are also very difficult to interpret. It might have been better to first explore the data using exploratory models and plots before deciding to fit all of the models. In any case, the total number of models fitted and the sample size needs to be quoted for each model, and the number of tests of multiple comparisons of means also needs to be recorded. What method of model selection / simplification was used? This should also be discussed.

R5. We thank the reviewer for pointing this out, and acknowledge that the difference between the two sets of models can indeed be hard to grasp since both explore the same response variables. To more clearly explain the difference between the two models to the reader, we now further clarified the hypotheses tested with each set of models in lines 239-251. In short, the first set of models tested for differences between the full set of treatments (that is, combinations and numbers of attackers) on plant performance (including treatments 1-15, with each treatment as a separate level), while the “additive models” tested for the effects of each of the attacker species on plant performance, and whether their combined effects on plant performance were additive or not (only including the seedlings from treatments 1-7, with the presence-absence of aphids, presence-absence of mildew, and presence-absence of caterpillars as predictors).

We agree that the four metrics of plant performance greatly increase the number of models fitted. Despite this, we still think it is important to inform the reader about all the plant performance metrics that were analyzed. As it is not unlikely that plant responses to attackers impact some traits but not others, multiple traits need to be tested to fully explore the potential effects of attackers on plant performance. Even in the absence of clear patterns, we would

argue it is still of value to report the lack of attacker impacts on these traits. However, to avoid overly many mentions of plant performance traits in the manuscript, we give the model outputs of these models and corresponding figures in the supplementary.

We do realize that the interpretation of three-way interactions can be particularly challenging. We now made sure to carefully explain the meaning of the three-way interactions in lines 261-263: *“Finally, the three-way interaction terms mildew × aphids × date, mildew × caterpillar × date and aphids × caterpillar × date tested whether the effects of the interactions between mildew and aphids, mildew and caterpillars, and aphids and caterpillars varied over time.”*

We now take care to include the sample sizes for all of the models in the tables of the supplementary materials and in the manuscript text where possible. We now mention the sample sizes for the plant performance models in the statistical section (lines 242 and 254), including the sample size for the each of the additive risk models (for which $n = 140$, which would be 840 observations, as each plant was measured weekly during 6 weeks).

Also, we added the number of tests of multiple comparisons of means in the Statistical analyses section of the Material and Methods (lines 285-286), and all pairwise comparisons (and statistical output) are reported in tables S7 and S8.

All models were decided upon *a priori* and were not further simplified or selected.

Line 230 square root, not squared-root

R6. Thanks for pointing out this glitch, we now corrected this.

General comment on figures: The ranges of the boxplots in the supp figures need to be explained. Boxplots should have a dot for means as well as medians shown, as medians are not always the best way to explore differences between treatment levels.

R7. Based on this comment and the comments of reviewer 2, we now changed the supplementary boxplots into bar graphs (see R33).

Line 290: But looking at figure 2a and as you state further down the results, isn't it the case that aphids only improved mildew performance when they were cooccurring, not when aphids arrived before mildew?

R8. The reviewer is correct that aphids only affected mildew when co-occurring, but not when aphids arrived before mildew. To better convey this message to the reader, we now rephrased lines 313-314 and line 325. We hope it is now clearer that in the paragraph in lines 312-322, we discuss the effects of co-occurring attackers on one another, whereas in the paragraph in lines 324-331 we go into the impact of early arriving attackers on later arriving attackers.

Lines 382-383: The suggestion that aphids induced the salicylic acid only locally...which pathway do they normally induce? My understanding is that biotrophic pathogens also induce the SA pathway, so the antagonistic hypothesis proposed would then make sense. Please clarify in the text which pathway you think each of the attackers induce.

R9. We realized that this sentence could create confusion about which pathway was expected to be induced. We now rephrased lines 409-410 to clarify that we expect an SA induction upon aphid arrival, but that this induction may be only local and not systemic, thereby potentially not affecting mildew in case the colonies are only present on distal plant tissues: *“For example, while the induction of SA is expected upon aphid attack, SA induction may occur only locally, i.e., at the feeding lesion (De Vos et al., 2005), which may explain the lack of a significant negative influence of aphids on mildew via SA-induction in our study.”*

Reviewer 2

In their paper « The timing and asymmetry of plant-pathogen-insect interactions », the authors address the effect of single vs. multiple attacks on plant growth as well as simultaneous vs. delayed interactions among attackers. They show that (1) pathogen and herbivore attack have weak or inconsistent effects on plant growth and that (2), when there are significant interactions among attackers, they are generally antagonistic and depend on the timing of the interaction. The most interesting result (for me) is that initial attack by a given attacker modifies the strength and direction of interactions between other attackers. The results are a bit complex and it is hard to navigate from the main text to the numerous supplementary materials, but I guess this is because word length limitation. The authors handled this fairly well. Overall, I liked the paper very much, because it is clear and well written, and because this is exactly the type of research I would have loved conducting. I only have a few comments/suggestions.

R10. We thank the reviewer for these kind words and the constructive feedback on our manuscript.

Paragraph starting L42 presents first the « positive » effect of attackers on plants, then the « negative » effects, but there is no clear transition between both, which makes it hard to follow. In addition, most of references are about herbivores. Any chance you can add references on pathogens as well.

R11. We agree that the stated contradiction about “positive to negative” effects on plant performance is not clearly followed up in the rest of the paragraph. To make sure the focus of the paragraph is clear (which is “variability in plant responses due to multiple mechanisms, such

as compensation”), we now removed the sub-sentence that stated that effects on plant performance “range from positive to negative” (lines 45-46).

We now made sure to include references on pathogens as well. The references by Fournier et al. (2006) and Hauser et al. (2013) are about the combined effect of pathogens and herbivores on plant performance. Schützendübel et al. (2008) discuss the susceptibility of barley to ramularia leaf spot disease depending on ontogenetic stage of the plant.

L64 « ...understand community dynamics » → AND resulting consequences on the host plant ?

R12. We now added this important consequence to lines 66-67.

L104 – This prediction is a bit vague (« synergistic or antagonistic »). Do you expect that some attackers would have more important consequences on plant growth ? Could you have a more precise prediction here ? Maybe not.

R13. Indeed, this prediction is based on the empirical evidence that suggests that effects can vary depending on the identity and combination of attackers, with synergistic as well as antagonistic impacts observed (Fournier *et al.*, 2006; Hauser *et al.*, 2013). To clarify to the reader what empirical evidence this hypothesis is based on, and how it is linked to the information we give earlier in the introduction, we now refer to Fournier et al. (2006) and Hauser et al. (2013) after stating this hypothesis. The follow-up sentence (lines 108-109) further explains that the impact of attackers on plant performance is expected to be dependent on the impact of attackers on each other. To make sure that it is clear to the reader that these predictions are linked, we now added “*Specifically*” (line 108) to the beginning of the follow-up sentence. Note that, as we did not *a priori* know how attackers would affect each other, we could not predict the expected direct of effects at this stage.

L125 (and elsewhere) is « family » really needed ?

R14. We agree and we now removed family for all species (lines 129, 133, 138).

L136 – It took me some time to realise that caterpillars may attack quite late in the season, after oak leaves have been colonized by mildew. This is because in other places, several defoliating species attack earlier in the season, at the same time as budburst. In this case, the interaction between mildew and caterpillars is more unidirectional. I think this could be more explicit here, for it can be confusing. Also, I wondered how common is *Bucephala* in the study area and whether the results would have been different with another species.

R15. The reviewer is correct in pointing out that oak is famous for high numbers of chewing caterpillars in the early season (Feeny, 1966). However, these high numbers of caterpillars in early spring are also known to be made up by a few highly abundant and heavily defoliating species. Later during the growing season there is a larger diversity of caterpillars feeding on oaks. Moreover, large-scale defoliation of oak trees in early spring is a rare phenomenon in central Sweden, compared with more southern latitudes. Caterpillars of *P. bucephala* frequently feed during the middle and end of the growing season, meaning the scenarios of time of arrival of the attackers simulated in this experiment are all likely to correspond to what often happens under natural conditions (see lines 153-155). To clarify the phenology of this species, we now added a statement to line 140, explaining that these caterpillars are feeding between July and early September.

P. bucephala is relatively common in southern and central Sweden, including the study county of Stockholm. Based on our hypothesis, that chewing insects generally induce the JA pathway, one might expect that the results would have been the same for other chewing caterpillars. However, there are also reasons to expect differences among caterpillar species. For example, the strength and type of induced defense responses, as well as the physiological response of the caterpillar to plant defense, can differ among caterpillar species, e.g. depending on their level of specialization (Ali & Agrawal, 2012). Moreover, as the reviewer also points out, some caterpillar species tend to arrive very early in the season, meaning some of our treatment scenarios will not be realistic for other species of caterpillars.

L140 – There are two cryptic mildew species, right ? How can you be sure yours is *E. alphitoides* ?

R16. While there are three cryptic (*Erysiphe alphitoides*, *Erysiphe hypophylla* and *Erysiphe quercicola*) and one non-cryptic mildew species found on oak in Europe, only two of the cryptic species are found in Sweden. These two species are easy to distinguish, as *E. alphitoides* is growing on the adaxial side of the leaf, whereas *E. hypophylla* is – as its Greek species name implies – strictly confined to the abaxial side of the leaf (Desprez-Loustau *et al.*, 2018). We have recently extensively sampled cryptic powdery mildew species and their hyperparasites on oak in Sweden. As in the previous studies, we found only *E. alphitoides* and *E. hypophylla*.

The interested reader can find this information in the references we now provide in this paragraph (Desprez-Loustau *et al.*, 2011, 2018), though we decided not to go into detail about the cryptic species in the manuscript to avoid dedicating overly many words to a topic that is not a main focus of the manuscript.

L152-158 – Any information about nutrients and watering treatment ?

R17. Thank you for pointing out this missing information. We now included a statement in this paragraph (lines 164-166) to clarify that seedlings had access to a continuous water supply

present in the tray in which the plant pot was standing, and that no additional nutrients were added to the potting soil.

L167 - « number of attackers » is unclear. This is the number of different attackers, not the density of a given attacker (that would have been another, really interesting question too).

R18. We now changed the text from “*number of attackers*” to “*number of attacker species*” to avoid confusion (line 176).

L169-175 – That would be nice to give the number of treatments for each of these five groups of treatments.

R19. We now specified the numbers that belong to each group of treatments in the main text (lines 181-185).

L181 – Were seedlings bagged individually ? If so, then I don’t understand the comment L206 (and following) suggesting that caterpillars were free to move across seedlings.

R20. We have now rephrased this sentence to make it clear that all seedlings were bagged individually and that caterpillars could not move across seedlings, but that caterpillars were removed after 1-2 weeks of feeding (line 218-219: “*Caterpillars were placed on a seedling for one to two weeks (depending on caterpillar size) after which they were removed (on seedlings from 9 treatments, thus 180 seedlings in total, see Fig. 1).*”).

L193 – Duplicated « the »

R21. We now corrected this error.

L194 – did you estimate the % leaf area covered by mildew or did you give an overall attack score at seedling level ? Is that representative of mildew infection ? What if some leaf area was missing because of herbivory ? How was it taken into account ?

R22. We thank the reviewer for pointing out some important missing details. We rephrased lines 204-206 to clarify that the estimation was at the seedling level (“*To obtain a measure of mildew infection at the seedling level, the percentage of the total leaf surface of a seedling that was covered by mildew was visually estimated each week by the same person (LJAvD).*”). Thus, this metric provided us with an overall estimate of the intensity of mildew infection on a seedling. Also, we now added a sentence to explain that leaf tissues eaten by caterpillars were not included in the mildew estimates, meaning that we estimated the percentage of mildew cover for the *remaining* leaf parts only (lines 207-208: “*In the presence of chewing herbivory, mildew coverage was estimated for the remaining leaf tissues only.*”). So, for example, if half of

the seedling was eaten by a caterpillar, and the remaining part of the seedling was covered in mildew for 50%, the estimated mildew coverage of the seedling would be 50% (and not 25%).

L217 – Because some caterpillars feed at night, or only during the day, the amount of food in the gut may vary with time, which means that there is a risk larva weight is not the same if weighed say at 9am or 4pm. Usually, people keep larvae for a few hours without food to avoid this bias. Not sure this is a big deal here. What do you think ?

R23. This is a good point. After careful consideration we would argue that, although it may have created more variation among individual caterpillar weights, it is unlikely that diurnal feeding patterns created any bias between treatments. We thus do not further discuss this issue in the manuscript. But still, this is indeed a very good point to be aware of in our future experiments.

L228 – Why was time fitted as a factor. There are 6 different levels. I would have used it as a continuous variable. The degrees of freedom would have been lower and it would have been easier to interpret the interaction term, as it would have been the slope. Here, with no indication on pairwise differences among time × treatment interactions, it is hard to figure out what it means.

R24. This is a good point. Our motivation for treating date as a factor was that this allows for testing also more complex, non-linear relationships, whereas with date as a continuous variable, growth is modelled as a linear process. To check if our choice to fit time as a factor rather than as a continuous variable influenced our results, we reran our models while treating date as a continuous variable. The output from these models was qualitatively the same as the output from models with date as a factor.

L242 – For additivity vs. non additivity, I had to write model equation to be convinced that the interaction term represents the non-additive effect. This is not absolutely needed but you may want to consider writing the equation. Just to be sure I understood properly, with e.g. A1 and A0 presence/absence of aphids, and M1/M0 presence/absence of mildew, and Y the predicted response variable:

$$Y = \beta_0 + \beta_1 \times A1 + \beta_2 \times M1 + \beta_3 \times A1 \times M1$$

with β_0 the intercept (corresponding to predictions for A0 and M0). If $\beta_3 = 0$ (i.e., no interaction), then Y in A1-M1 is simply predicted from A1 only, M1 only and $\beta_0, \beta_1, \beta_2$, is that correct?

I realized that models are provided in supplementary tables. I am fine with the notation, but

some people complain that what you report is not model equation, strictly speaking. There is nothing to do with this comment, just keep in mind that sometimes it helps having the model equation(s) properly written.

R25. We agree with the reviewer that this model is particularly challenging to wrap one's head around, which is also why we backed up this approach with references to previous studies taking the same approach. That said, the reviewer indeed interpreted the model correctly: If the interaction term is significant, the response of the plant is higher or lower than one would have expected from mildew and aphids alone. Thanks for this notation, we will keep this in mind.

L252 – When analyzing growth rate, it is usually suggested to use initial weight as a covariate and to interpret the initial weight \times treatment interaction as growth rate. Here, you only mention « final instar weight », while initial weight was variable across individuals.

R26. Thank you for pointing this out. We now modelled both final caterpillar weight and pupal weight as a function of the categorical, fixed effect treatment and the continuous, fixed effect initial weight. While the outcome did not differ qualitatively (lines 321-322), we agree that it is important to account for initial weight in the analysis.

Discussion

L273 – Can you precise whether the effect of acorn size was positive or negative ?

R27. Thank you for pointing out this missing information, the text has now been changed from “effects of acorn size” to “positive effects of acorn size” (line 296).

L298 – Treatment, not diet

R28. We now changed this to “treatment” (line 321).

L331 – You only focused on above ground growth. It is possible that treatments had a strong influence on belowground growth. This is particularly important for seedlings. Also, the effects of treatments on oak seedlings may have been delayed. Is there any chance you looked at growth the next year ?

R29. We agree with the reviewer that both belowground and long-term effects present very interesting research avenues. Unfortunately, we did not pursue these factors in this experiment and are therefore not able to address these questions. Another variable that could be explored in long term experiments is the impact of ontogenetic stage of the seedling, and how age matters for the plants' response towards attackers. To highlight these interesting future

research avenues in our manuscript, we now elaborated the discussion on these topics (lines 377-378, lines 390-391).

Also, about growth, is it possible that treatments were not « damaging » enough ? I would have reported mildew infection, % herbivory and aphid load in the 'Results' section and discuss whether these scores are high/low and representative or not of what could be seen in the wild.

R30. While we realize that many studies expose their plants to very (often unnaturally) high intensities of damage to increase the likelihood of a significant effect, we deliberately strived to use naturally realistic attacker densities. To clarify this, we now added a statement in the material and methods, stating that our experimental observations are likely to resemble natural conditions (lines 156-158: *“Furthermore, we strived to expose our seedlings to realistic attacker densities corresponding to the levels of attack and damage experienced by oak seedlings under natural conditions (Tack et al., 2010; Ekholm et al., 2017; Faticov et al., 2020).”*).

We took care that information about population sizes for all attackers is present in the manuscript: The reader can get an idea of the average aphid population sizes / mildew infection rates from Figures 2-3, and we discuss the average damage from caterpillars in the material and methods in line 223. Note that there was considerable variation in attacker densities among treatments and seedlings, which makes it difficult to discuss attacker densities more generally.

L379 – was the effect of aphid « wound » on mildew really relevant ? I suspect that the mechanical effect of leaf chewing by caterpillars is more significative.

R31. The reviewer is correct in pointing out that this statement is quite speculative. While the impact of puncture damage on fungal colonization has been described for necrotrophic fungi and obligate fungal pathogens that grow their mycelium intercellularly (e.g. rust fungi; Hatcher (1995), the impact of puncture damage on epiphytically-growing plant pathogens that only enter the epidermal cells with their feeding organs (e.g. powdery mildews) remains elusive. So, upon further consideration, we decided to remove this speculative mechanism from the discussion, and replaced it with an explanation that is more solidly based on empirical evidence (lines 403-405: *“For example, aphids can act as vectors for fungal spores (Kluth et al., 2002), and may thus aid in the spread of disease within a seedling.”*).

L385 – is there any chance you looked at interactions at the level of individual leaves ? Were interactions local or only distant ? That would help discussing the effect of local vs systemic defense induction

R32. That is a very interesting angle indeed, but unfortunately our experimental setup was not designed to distinguish between local and distant interactions on the seedling level. To clarify this, we now modified our statement in lines 412-413 in the discussion (*“While it was not the*

aim of our study to distinguish local from distant interactions, we encourage future studies to explore the effect of contemporaneous SA and JA induction in nearby and distant plant tissues, and thereby resolve how spatial scale affects defence crosstalk (Spoel et al., 2007) and shapes attacker interactions.”).

Figure 4 helps a lot. But : (1) could you make arrow width proportional to model coefficient (1bis) avoid grey vs. red; and (2) I like cartoons but I am not a big fan of having realistic vs. cartoonish drawings in the same figure.

Figures – That would be nice to have the results of contrast analysis (where appropriate) indicating which comparisons are significant or not. Also, I did not really understand why in some instance you have bar graphs and, elsewhere, boxplots or lines. OK, lines better show some continuity in the response variable (e.g., seedling measurements), but then, you could have used it for mildew coverage and aphid population size as well, for the same reason.

R33. Thanks for these suggestions to improve our figures.

On figure 4:

(1) While we think it is very tempting to make arrow widths proportional to model coefficients, we were hesitant to implement this because the results in this figure originated from different models, and model outputs are not directly comparable. We thus fear that displaying arrow widths based on coefficients from different models may be misleading. As suggested, we now replaced grey arrows by white arrows with black lining.

(2) We fully agree with the reviewer that the cartoons did not have the same level of realism. We now made new cartoons for all study organisms, and implemented these in all figures that display cartoons (Figs 1, 2, 3, 4, S2 and S4).

On figures 2-3: We now included letters to indicate significance within the figures of the main manuscript.

On all figures: To be more consistent in our use of figures, we replaced the boxplots by bar graphs (see also R7 to Reviewer 1).

It is a bit frustrating the title does not reflect very well the fact that you also address the effect of single vs. multiple attacks on the plant.

R34. We agree that this part of the study is not well reflected in the title. To make sure we emphasize this part of our study already early on in the manuscript, we now added a sub-sentence to the abstract (lines 17-18), explaining that we also looked into the effect of single vs. multiple attackers on plant performance. (*“In this study, we assessed the impact of time of attacker arrival on the outcome and symmetry of interactions between aphids (Tuberculatus annulatus), powdery mildew (Erysiphe albitoides) and caterpillars (Phalera bucephala) feeding*

on pedunculate oak, Quercus robur, and explored how single vs. multiple attackers affect oak performance.”)

Otherwise, this is a very nice paper :)

References

- Ali JG, Agrawal AA. 2012.** Specialist versus generalist insect herbivores and plant defense. *Trends in Plant Science* **17**: 293–302.
- De Vos M, Van Oosten VR, Van Poecke RMP, Van Pelt JA, Pozo MJ, Mueller MJ, Buchala AJ, Métraux J-P, Van Loon LC, Dicke M, et al. 2005.** Signal Signature and Transcriptome Changes of Arabidopsis During Pathogen and Insect Attack. *Molecular Plant-Microbe Interactions* **18**: 923–937.
- Desprez-Loustau M-L, Feau N, Mougou-Hamdane A, Dutech C. 2011.** Interspecific and intraspecific diversity in oak powdery mildews in Europe: coevolution history and adaptation to their hosts. *Mycoscience* **52**: 165–173.
- Desprez-Loustau M-L, Massot M, Toïgo M, Fort T, Aday Kaya AG, Boberg J, Braun U, Capdevielle X, Cech T, Chandelier A, et al. 2018.** From leaf to continent: The multi-scale distribution of an invasive cryptic pathogen complex on oak. *Fungal Ecology* **36**: 39–50.
- Ekholm A, Roslin T, Pulkkinen P, Tack AJM. 2017.** Dispersal, host genotype and environment shape the spatial dynamics of a parasite in the wild. *Ecology* **98**: 2574–2584.
- Faticov M, Ekholm A, Roslin T, Tack AJM. 2020.** Climate and host genotype jointly shape tree phenology, disease levels and insect attacks. *Oikos* **129**: 391–401.
- Feeny PP. 1966.** *Some effects on oak-feeding insects of seasonal changes in the nature of their food.* University of Oxford.
- Fournier V, Rosenheim JA, Brodeur J, Diez JM, Johnson MW. 2006.** Multiple Plant Exploiters on a Shared Host: Testing for Nonadditive Effects on Plant Performance. *Ecological Applications* **16**: 2382–2398.
- Hatcher PE. 1995.** Three-way interactions between plant pathogenic fungi, herbivorous insects and their host plants. *Biological Reviews* **70**: 639–694.
- Hauser TP, Christensen S, Heimes C, Kiær LP. 2013.** Combined effects of arthropod herbivores and phytopathogens on plant performance. *Functional Ecology* **27**: 623–632.
- Kluth S, Kruess A, Tschardt T. 2002.** Insects as Vectors of Plant Pathogens: Mutualistic and Antagonistic Interactions. *Oecologia* **133**: 193–199.
- Mursinoff S, Tack AJM. 2017.** Spatial variation in soil biota mediates plant adaptation to a foliar pathogen. *New Phytologist* **214**: 644–654.
- Nicot PC, Bardin M, Dik AJ. 2002.** Basic methods for epidemiological studies of powdery mildews: Culture and preservation of isolates, production and delivery of inoculum, and disease

assessment. In: *The powdery mildews: A comprehensive treatise*. St. Paul, Minnesota: The American Phytopathological Society, 83–100.

Schützendübel A, Stadler M, Wallner D, Tiedemann AV. 2008. A hypothesis on physiological alterations during plant ontogenesis governing susceptibility of winter barley to ramularia leaf spot. *Plant Pathology* **57**: 518–526.

Spoel SH, Johnson JS, Dong X. 2007. Regulation of tradeoffs between plant defenses against pathogens with different lifestyles. *Proceedings of the National Academy of Sciences* **104**: 18842–18847.

Tack AJM, Ovaskainen O, Pulkkinen P, Roslin T. 2010. Spatial location dominates over host plant genotype in structuring an herbivore community. *Ecology* **91**: 2660–2672.